# Antisense Morpholino-Based In Vitro Correction of a Pseudoexon-Generating Variant in the *SGCB* Gene

**DOI:** 10.3390/ijms23179817

**Published:** 2022-08-29

**Authors:** Francesca Magri, Simona Zanotti, Sabrina Salani, Francesco Fortunato, Patrizia Ciscato, Simonetta Gerevini, Lorenzo Maggi, Monica Sciacco, Maurizio Moggio, Stefania Corti, Nereo Bresolin, Giacomo Pietro Comi, Dario Ronchi

**Affiliations:** 1Neurology Unit, IRCCS Fondazione Ca’ Granda Ospedale Maggiore Policlinico, 20122 Milan, Italy; 2Neuromuscular and Rare Disease Unit, IRCCS Fondazione Ca’ Granda Ospedale Maggiore Policlinico, 20122 Milan, Italy; 3Dino Ferrari Center, Department of Pathophysiology and Transplantation, University of Milan, 20122 Milan, Italy; 4Unit of Neuroradiology, Papa Giovanni XXIII Hospital, 24127 Bergamo, Italy; 5Neuroimmunology and Neuromuscular Diseases Unit, Fondazione IRCCS Istituto Neurologico Carlo Besta, 20133 Milan, Italy

**Keywords:** LGMD, *SGCB*, beta-sarcoglycan, morpholino

## Abstract

Limb-girdle muscular dystrophies (LGMD) are clinically and genetically heterogenous presentations displaying predominantly proximal muscle weakness due to the loss of skeletal muscle fibers. Beta-sarcoglycanopathy (LGMDR4) results from biallelic molecular defects in *SGCB* and features pediatric onset with limb-girdle involvement, often complicated by respiratory and heart dysfunction. Here we describe a patient who presented at the age of 12 years reporting high creatine kinase levels and onset of cramps after strenuous exercise. Instrumental investigations, including a muscle biopsy, pointed towards a diagnosis of beta-sarcoglycanopathy. NGS panel sequencing identified two variants in the *SGCB* gene, one of which (c.243+1548T>C) was found to promote the inclusion of a pseudoexon between exons 2 and 3 in the *SGCB* transcript. Interestingly, we detected the same genotype in a previously reported LGMDR4 patient, deceased more than twenty years ago, who had escaped molecular diagnosis so far. After the delivery of morpholino oligomers targeting the pseudoexon in patient-specific induced pluripotent stem cells, we observed the correction of the physiological splicing and partial restoration of protein levels. Our findings prompt the analysis of the c.243+1548T>C variant in suspected LGMDR4 patients, especially those harbouring monoallelic *SGCB* variants, and provide a further example of the efficacy of antisense technology for the correction of molecular defects resulting in splicing abnormalities.

## 1. Introduction

Limb-girdle muscular dystrophies (LGMD) are hereditary disorders characterized by the loss of skeletal muscle fibers, resulting in predominantly proximal muscle weakness at onset. The impressive genetic heterogeneity is acknowledged by the report of mutations in more than 30 genes, displaying dominant and recessive patterns of inheritance, associated with LGMD presentation [1].

Biallelic variants in *SGCA*, *SGCB*, *SGCG*, and *SGCD* encoding for alpha-, beta-, gamma-, and delta-sarcoglycan proteins [2], respectively, are the molecular determinants of rare recessive LGMD forms, collectively termed sarcoglycanopathies (LGMDR3-6, according to the updated nomenclature) [3]. Sarcoglycans interact with the dystroglycan complex, which links the subsarcolemmal protein dystrophin to the basement membrane. In this way, sarcoglycans participate in the maintenance of muscle membrane integrity during muscle fibers’ contraction and relaxation process [4]. Indeed, molecular defects in any of the four sarcoglycan genes result in the disruption of the whole complex and the loss of sarcolemmal integrity.

Pathogenic variants in *SGCB*, encoding for beta-sarcoglycan, result in the recessive limb-girdle muscular dystrophy-4 (LGMDR4) form, displaying progressive weakening of proximal muscles in early childhood [5]. Compared to other LGMDs, LGMDR4 displays an earlier age of onset (6.4 ± 5.2 years according to the Italian national registry and 5.8 ± 4.1 years according to the European Sarcoglycanopathy Consortium) [6,7], although few patients presenting in the second decade have been also reported. Muscle weakness predominates in proximal muscles, with a more severe involvement of lower limbs, like other LGMDs. Axial and distal involvement is frequent and prominent in severe forms, even at the early stages of disease [7].

Respiratory and heart dysfunction resemble those observed in patients with Duchenne Muscular Dystrophy [8]. Dilated cardiomyopathy has been observed in more than 50% of LGMDR4 patients, while it is less common in other sarcoglycanopathies. Progressive respiratory failure, if not prevented using non-invasive ventilation, constitutes a major cause of death in a relevant number of LGMDR4 patients.

A muscle biopsy is still an informative tool for the diagnosis of LGMD patients. The loss of sarcoglycan signal in immunohistochemistry followed by confirmative quantitative Western blot experiments drives the molecular analysis. The combined use of sequencing (traditional or next-gen approaches) and quantitative techniques (such as MLPA) allows for the identification of causative variants in almost all patients with a suspect of sarcoglycanopathy [6]. Molecular defects in *SGCB* are distributed along the whole gene, but the 8-bp c.377_384dup duplication in exon 3 accounts for 20% of the mutated alleles [9]. A small number of LGMDR4 cases escape molecular diagnosis, suggesting the existence of molecular defects in regions that are not usually addressed by molecular screening [10].

Here we report a novel LGMDR4 patient presenting as almost asymptomatic at 12 years of age and with repeated findings of high creatine-kinase levels. A muscle MRI pointed towards a diagnosis of LGMD and a muscle biopsy showed a severe reduction in sarcoglycans. NGS-based analysis followed by *SGCB* cDNA analysis allowed the identification and characterization of a deep intronic variant influencing *SGCB* splicing. The in vitro administration of an antisense morpholino oligomer partially rescued the splicing defect in the patient’s induced pluripotent stem cells (iPSCs).

## 2. Results

### 2.1. Case Report

The patient is now a 16-year-old boy who had sought medical attention at 12 years of age because of the incidental finding of high creatine-kinase levels (8413 IU/L) confirmed at further analysis.

He was a healthy boy, born from non-consanguineous parents after regular pregnancy and delivery and with normal psychomotor development. The patient played soccer at a competitive level and was asymptomatic; he only complained of cramps after strenuous exercise. He did not show weakness, fatigability, myalgias, or myoglobinuria. A neurological examination at 12 years of age was normal except for mild sural pseudohypertrophy. A cardiological evaluation did not show abnormalities. A muscle MRI performed at 13 years of age only showed mild T2 hyperintensity in right triceps surae.

A muscle biopsy, performed on the brachial biceps at 13 years of age, showed mild myopathic signs (few hypotrophic fibers and two necrotic fibers, Figure 1A). The IHC study was normal, except for staining with antibodies against alpha- (Figure 1B) and gamma-sarcoglycan (Figure 1C), which revealed a severe reduction in reactivity in many fibers. A severe (>70%), but not total, reduction in alpha-, beta-, gamma-, and delta-sarcoglycans was also confirmed by Western blot analysis (Figure 1D) with residual protein levels of 25%, 29%, 27%, and 29%, respectively, compared to control biopsies.

The patient was evaluated annually for the subsequent four years, showing mild progression of the disease. A muscle MRI was repeated at 14 years of age and showed mild fibroadipose substitution of the glutei, left paraspinal muscles, serratum, and teres major. Inflammatory signs were absent.

At 16 years of age, the patient was still asymptomatic and able to play soccer, complaining only of cramps. Neurological examination showed a normal strength, evaluated through Medical Research Council (MRC) Scale with the exception of the ileopsoas (4.5 at MRC evaluation). At functional evaluation, the patient showed a reduced distance walked during a 6 min walking test (525 m compared to 578 m walked at 12 years of age). At Upper Limb Function scale, Motor Function Measure scale, and adjusted North Star scale, the patient reached the best score available at both ages [11,12,13].

At 16 years of age, electrocardiography was normal, but echocardiography showed a mild initial left ventricular hypertrophy with an ejection fraction of 54%. Respiratory function was normal.

### 2.2. Genetic Studies

NGS panel sequencing prioritized the heterozygous *SGCB* variant chr4:528958890/TCCTACTG corresponding to the microduplication c.377_384dup (NM_000232) (Figure 2A and Appendix A). This change is classified as a class 5 (pathogenic) variant according ACGM guidelines and results in a premature termination codon (p.Gly129Glnfs*2). This variant has been previously described in several LGMDR4 patients [7]. No other variant was detected in the *SCGB* coding sequence. MLPA analysis ruled out *SGCB* deletions or duplications in the proband and his parents. The re-examination of aligned readings allowed us to detect the ultrarare (single occurrence in GnomAD database) heterozygous substitution chr4:52898049A/G (Appendix A) in a deep-intronic region between exons 2 and 3 (c.243+1548T>C), belonging to a predicted (ENST00000506357) pseudoexon (PE) of 110 base pairs (Appendix A). The c.243+1548T>C substitution, classified as a class 3 variant (variants of uncertain significance) according ACGM guidelines, is predicted to alter an exonic splicing enhancer (ESE) recognized by SRSF2 splicing factor (Appendix A), favoring the PE’s inclusion in the mature transcript. Segregation analysis confirmed that the *SGCB* variants c.377_384dup and c.243+1548T>C were inherited from the unaffected patient’s father and mother, respectively (Figure 2B). The CK levels in the patient’s parents were normal.

We re-examined a DNA sample from a patient deceased more than 20 years ago ([10] Patient 1, V-5), whose clinical information at onset partially overlapped with that observed in our proband. A muscle biopsy had revealed a moderate reduction in beta-sarcoglycan protein, but molecular analysis had only detected the c.377_384dup, inherited from the unaffected father. Interestingly, Sanger sequencing revealed the same c.243+1548T>C variant in this patient and his unaffected mother (Appendix A). An updated segregation of *SGCB* variants in this second pedigree is now provided in Appendix A.

We analyzed microsatellite markers upstream and downstream of the *SGCB* locus in available DNA samples. A haplotype of the allele harboring the c.377_384dup was conserved between the two pedigrees and other independent *SGCB* patients of our cohort displaying the same defect, providing support to the hypothesis of the existence of a founder allele in LGMRD4 patients from Northern Italy carrying the c.377_384dup variant [14]. Interestingly, although familiarity between the two family groups was denied, the allele harboring the c.243+1548T>C variant also displayed the identical microsatellite haplotype in the affected probands (Appendix A), advancing the hypothesis that the affected patients are distantly related.

### 2.3. Transcript Analysis

To evaluate the consequence of the c.243+1548T>C variant, we analyzed the *SGCB* transcript after retrotranscription of muscle-extracted RNA. RT-PCR amplicons encompassing exons 2 and 3 showed faint bands corresponding to physiological splicing products and the presence of additional bands with higher molecular weight in the patient’s muscles compared to controls (Figure 2C). Sequence analysis confirmed the abnormal inclusion of the 110 bp pseudoexon between exon 2 and exon 3 in these alternative splicing products (Figure 2D). We designed quantitative RT-PCR assays to detect *SGCB* transcript levels. In the patient’s muscles, levels of transcripts including Exon 4–5 junction were reduced compared to controls, likely as the effect of the frameshift c.377_384dup defect. Levels of transcripts including the Exon 2–3 junction were further reduced, reflecting the existence of alternative splicing products (Figure 2E). Indeed, higher levels of transcripts containing the Exon 2-PE junction were observed in the patient’s muscles compared to controls (Figure 2E,F).

We extended these studies to muscle biopsy and autoptic heart specimens from the previously described LGMDR4 patient sharing the *SGCB* genotype of our patient. Qualitative and quantitative experiments documented the same changes (Appendix A). Automated high-performance electrophoresis (TapeStation analysis) in available specimens also showed the presence of aberrant splicing products and a low amount of normal RT-PCR amplicons in mutated samples. Interestingly, few molecules of PE-containing transcripts were observed in control samples (Appendix A).

### 2.4. In Vitro Delivery of Antisense Morpholino Oligomer

Previous findings suggest that the c.243+1548T>C variant favors the inclusion of PE in *SGCB* transcripts, an event that occurs physiologically in a low amount of mRNA molecules. Morpholino-based antisense oligomers are among the most promising strategies to modulate alternatively spliced products and have been previously tested as therapeutic interventions for several splicing defects [15]. A morpholino sequence (MO) was designed to target the c.243+1548T>C variant and the downstream splice site junction aiming to PE skipping (Figure 3A). Since the patient’s primary cells were unavailable, we generated induced pluripotent stem cells (iPSCs) starting from the patient’s lymphocytes. In vitro delivery of 10 µM MO in the patient’s iPSCs cells for 48 and 72 h resulted in the complete correction of alternative splicing due to PE inclusion and a robust increase in physiological splicing products compared to untreated cells (Figure 3B,C). RT-PCR analysis followed by direct sequencing confirmed these findings. No rescue was observed following the treatment with a morpholino scrambled (SCR) sequence (Appendix A).

Immunocytochemical detection of the SGCB protein displayed a progressive increase in beta-sarcoglycan signal in MO-treated cells for 48 and 72 h compared to untreated iPSCs (Figure 3D).

## 3. Discussion

Our study expands the genetic findings linked with beta-sarcoglicanopathies by identifying and characterizing a pseudoexon-generating *SGCB* variant in a novel LGMDR4 patient as well as in a previously reported subject, still lacking a molecular diagnosis. The availability of patients’ tissues allowed the identification of the splicing defect leading to beta-sarcoglycan deficiency. The establishment of patient-specific induced pluripotent stem cells provided the work bench to test a morpholino oligomer aiming at restoring the physiological splicing.

Our patient presented at 12 years of age with oligosymptomatic hyperckemia. Despite the mild phenotype observed at onset, he has recently developed mild lower limb proximal weakness and left ventricular hypertrophy, as frequently observed in other cases of beta-sarcoglycanopathy. These data highlight the importance of a frequent follow-up of LGMDR4 patients even in the pre-symptomatic or pauci-symptomatic stage of the disease.

About 60 different *SGCB* molecular defects have been associated with LGMDR4, missense, non-sense, and indels being the most common pathogenic variants. Splicing-affecting variants have rarely been reported [2].

Our proband displayed the c.377_384dup, a detrimental molecular defect usually leading to severe phenotype with absence of the sarcoglycan proteins in skeletal muscle when present in homozygosis or in trans with a different null mutation [9]. The extension of next-generation panel sequencing to non-canonical coding exons of *SGCB* resulted in the identification of the variant c.243+1548T>C. Several elements support the pathogenicity of this defect: (i) it displays an extremely low frequency in the population database; (ii) it was detected in an independent patient with overlapping clinical features at onset; (iii) it was found in trans with respect to c.377_384dup, confirming the biparental inheritance and matching the recessive behavior of *SGCB* defects; (iv) transcript analysis in available tissues (skeletal, muscle, and heart) from mutated patients showed the same effect on *SGCB* mRNA; (v) the documented pseudoexon inclusion supports the in silico prediction of the loss of an ESE sequence in presence of the c.243+1548T>C transition. We believe that the genomic region hosting this pseudoexon should be included in the molecular screening of LGMDR4 patients, especially those presenting a single *SGCB* pathogenic variant with moderate or severe sarcoglycans deficiency in muscle.

Our study provides a conclusive molecular diagnosis for the patient reported by Barresi et al. and deceased more than twenty years ago [10]. That patient developed dilated cardiomyopathy at 23 years of age and died of acute cardiac insufficiency at the age of 26. In addition, he developed skeletal muscle weakness leading to the loss of independent ambulation at the age of 23 years.

Cardiac involvement is frequent in LGMDR4; more than 60% of LGMDR4 patients suffer from dilated cardiomyopathy [16], with onset of cardiac involvement at around 25 years of age [6,7]. A recent study confirmed the higher prevalence of dilated cardiomyopathy in LGMDR4 presenting mono- or biallelic *SGCB* c.377_384dup [17], strengthening the existence of a phenotype–genotype correlation. In our patient, initial cardiac involvement was observed at the age of 16 years. A close follow-up will be required to monitor cardiac function, and eventually, to establish a pharmacological regimen to slow down the progressive heart disease, which proved fatal in the other patient with the same genotype [10].

By analyzing tissues collected from the two patients presenting the c.243+1548T>C transition, we demonstrated the existence of low levels of normally spliced *SGCB* transcripts. This is compatible with the residual amount of beta- and other sarcoglycan proteins detected in Western blot analysis. Residual protein levels of sarcoglycans generally correlate with age at onset [6] and age of independent ambulation loss [7]. LGMDR4 patients displaying total absence of beta-sarcoglycan in muscle present an earlier onset than patients with residual protein expression (3.1 ± 1.4 years vs. 13.7 ± 5.7 years) [6], and patients with muscle protein levels higher than 30% have a lower risk to lose independent ambulation before 18 years of age [7]. No correlation between muscle protein levels and cardiac involvement has been detected so far in LGMDR4 patients. All sarcoglycans are robustly expressed in skeletal muscle [2], while their expression in cardiac muscle is unbalanced with higher levels of SGCB, SGCG, and SGCD compared to SGCA. Results from Northern blot experiments also hypothesized tissue specific SGCB isoforms [5].

Antisense oligonucleotide technology has been extensively used to modulate the expression of the target mRNA sequence. Splice modulation is achieved by introducing chemical modifications to avoid RNase-H degradation of the oligo-RNA hybrid [18]. Morpholino-based antisense oligomers are a best-in-class technology for the therapeutic modulation of splicing, and their use is now approved to treat Duchenne Muscular Dystrophy caused by specific defects in the dystrophin gene [19].

To our knowledge, this study represents the first application of MO to an *SGCB* variant. In our study, the delivery of Morpholino oligomers targeting the c.243+1548T>C variant to the patient’s iPSCs resulted in the complete correction of aberrant splicing and a robust increase in normal *SGCB* transcripts compared to untreated cells. Increased expression of beta-sarcoglycan was also observed in MO-treated cells. Our findings constitute a proof of principle evidence of the use of morpholino to bypass the pseudoexon included by c.243+1548T>C variant in the *SGCB* transcript and add up to similar results previously obtained in the dysferlin gene by independent investigators [20].

No therapy is available for LGMDR4. Promising results in preclinical studies addressing the efficacy of systemic AAV-mediated beta-sarcoglycan delivery [21] prompted the forthcoming phase I/II study (NCT03652259). Antisense oligonucleotides constitute an alternative therapeutic approach that can be tailored to rescue specific and private splicing defects, avoiding risks and limitations of gene-replacement strategies. Morpholino antisense technology has been already used to bypass exons containing defects in *SGCG*, encoding for gamma-sarcoglycan, and was found to rescue the function of the sarcoglycan complex [22]. The application of a similar strategy to in-frame coding exons of *SGCB* could further expand the repertoire of *SGCB* variants amenable to treatment with morpholino oligomers.

## 4. Materials and Methods

### 4.1. Muscle Tissue Analysis

The proband underwent a muscle biopsy after giving written informed consent, according to Institutional guidelines. Morphological examination was performed according to standard procedures.

Immunohistochemical (IHC) analyses were performed using monoclonal antibodies directed against three different epitopes of dystrophin (rod-domain diluted 1:10, NH_2_-domain diluted 1:10, COOH-domain undiluted, all from Novocastra, Newcastle upon Tyne, UK), sarcoglycans (alpha-sarcoglycan 1:20, gamma-sarcoglycan 1:10, Novocastra, Newcastle upon Tyne, UK), and caveolin-3 (1:1000, BD Transduction Laboratories, Franklin Lakes, NJ, USA) as previously described [23].

Forty micrograms of muscle protein lysates was probed for calpain-3 (Novocastra, 2C4, 1:100), dysferlin (Novocastra, Newcastle upon Tyne, UK, Hamplet-2 1:1000), dystrophin (rod-domain 1:200 and COOH-term 1:80), actinin (Sigma, St. Louis, MO, USA, 1:10,000), and sarcoglycans (alpha-sarcoglycan 1:280, beta-sarcoglycan 1:65, gamma-sarcoglycan 1:200, delta-sarcoglycan 1:60, all from Novocastra) expressions by Western blot (WB) analysis on a 10% polyacrylamide gel.

### 4.2. Molecular Studies

Genomic DNA was extracted from peripheral blood samples according to standard procedures. A custom panel including 56 genes associated with muscle disorders (Appendix A) and covering all relevant genes associated with LGMDs was investigated by using a 150 bp amplicon-based approach (Haloplex Target Enrichment System, Agilent, Santa Clara, CA, USA). Libraries were sequenced on an MiSeq instrument (Illumina, San Diego, CA, USA). Readings were aligned to the human genome (assembly hg19), and the identified variants were annotated (ANNOVAR) and filtered, focusing on rare variants (≤0.5% in public databases) causing changes potentially damaging for the protein function (CADD and DANN). *SGCB* variants were re-sequenced for confirmation and segregation studies by Sanger analysis on an ABI Prism 3130 platform (Applied Biosystems, Waltham, MA, USA). *SGCB* locus microsatellite markers were also analyzed by capillary electrophoresis on the same instrument.

mRNA was isolated from tissues and cells with Eurozol. Then, cDNA was produced through a reverse transcription-polymerase chain reaction (RT-PCR) using the RT Maxima Reverse Transcription Master Mix (Thermo Fisher, Waltham, MA, USA). RT-PCR amplicons were electrophoresed on agarose gels and on the Agilent Tape Station 2100 and therefore analyzed by Sanger sequencing. Quantitative RT-PCR experiments were performed in triplicate by SYBR green-based chemistry on 7500 Real Time PCR Instruments (Applied Biosystems, Waltham, MA, USA). Primers are available upon request.

### 4.3. Cell Cultures

The patient’s lymphocytes were reprogrammed to induced pluripotent stem cells (iPSCs) using the CytoTune-iPS 2.0 Sendai Reprogramming Kit (Thermo Fisher, Waltham, MA, USA). The morpholino (MO) sequence (5′-CGGGACTTGCCAAATGACGGTATCT-3′) and the nonspecific scrambled (SCR) sequence (5′-GTATTGGATTGCGCCACACCGTGAA-3′) were designed and synthesized as Bare-MOs without any modifications or with an octa-guanidine modification by Gene Tools (Philomath, OR, USA). MOs were dissolved in sterile water and delivered at a final concentration of 10 µM.

## Figures and Tables

**Figure 1 ijms-23-09817-f001:**
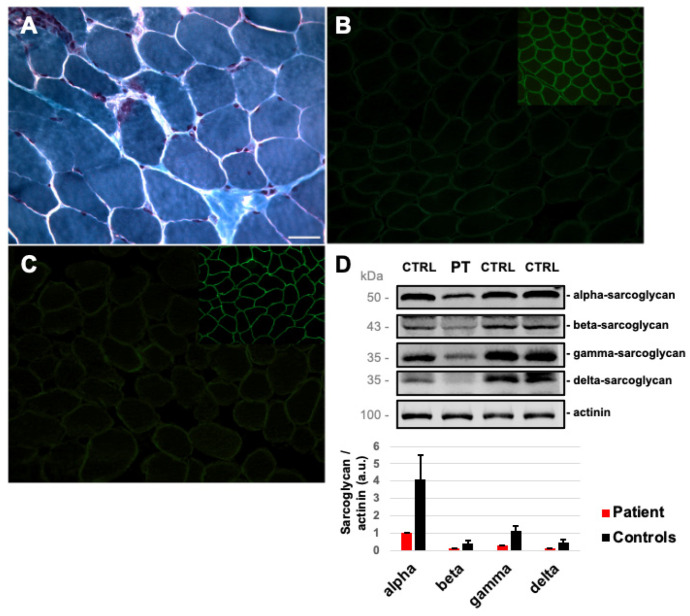
Skeletal muscle involvement in our Patient: (**A**) Modified Gomori Trichrome on patient’s muscle section showing the presence of necrotic fibers. Immunofluorescent staining for alpha-sarcoglycan (**B**) and gamma-sarcoglycan (**C**) (insert: alpha- and gamma-sarcoglycan in control muscle sections). Magnification 20×. Scale bar 50 μm. (**D**) Western blot analysis of sarcoglycans proteins and muscle actinin used as a reference in patient’s (PT) and control (CTRL) muscle biopsies. Protein levels of sarcoglycans normalized to actinin are reported in the bar chart (error bars mean standard deviations among controls, *n* = 3).

**Figure 2 ijms-23-09817-f002:**
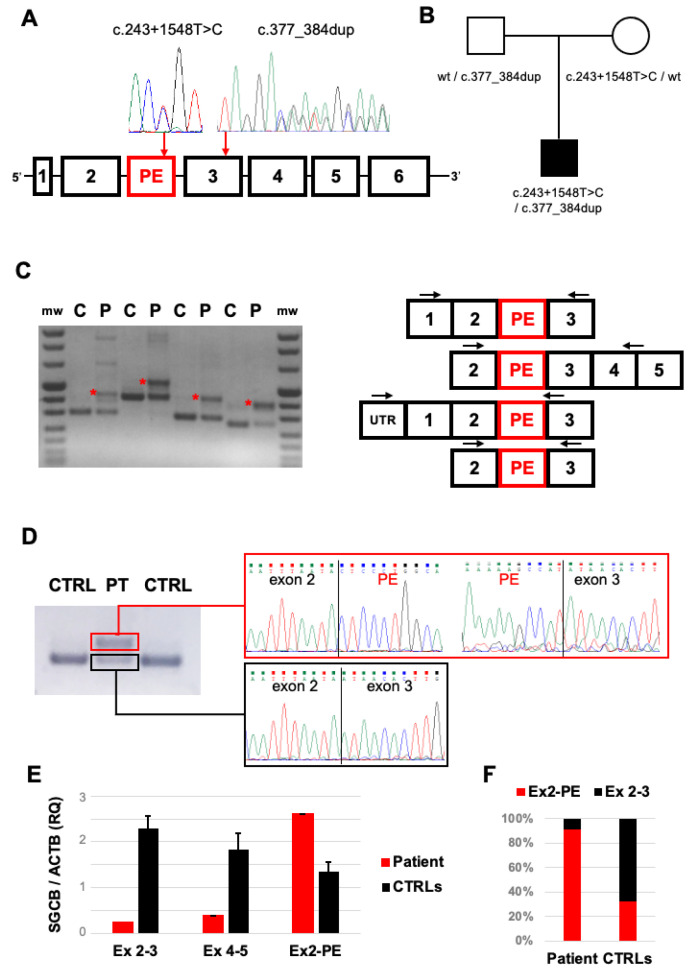
Molecular studies: (**A**) Scheme of *SGCB* gene displaying the position of the predicted pseudoexon (PE) and sequence electropherograms of heterozygous c.243+1548T>C and c.377_384dup variants identified in our patient. (**B**) Segregation analysis in the trio described in the family supporting biallelic inheritance of the *SGCB* variants. (**C**) RT-PCR analysis of *SGCB* transcripts in patient’s (P) and control (C) muscle specimens. The scheme of multiple PCR amplicons encompassing the cDNA region containing the PE is reported (arrows indicate the positions of PCR primers). (**D**) Sequence electropherograms generated from RT-PCR amplicons displaying the abnormal Exon2-PE-Exon3 junction in patient’s (P), but not control (C), muscle cDNA. (**E**) Quantitative RT-PCR experiments evaluating physiological (Ex2–Ex3, Ex4–Ex5) and abnormal (Ex2-PE) splicing junctions in muscle cDNA retrotranscribed from patient’s muscle and 3 control muscle biopsies. (**F**) Percentage of pseudoexon inclusion (Ex2-PE, red) in patient’s and control muscle biopsies estimated by qRT-PCR experiments.

**Figure 3 ijms-23-09817-f003:**
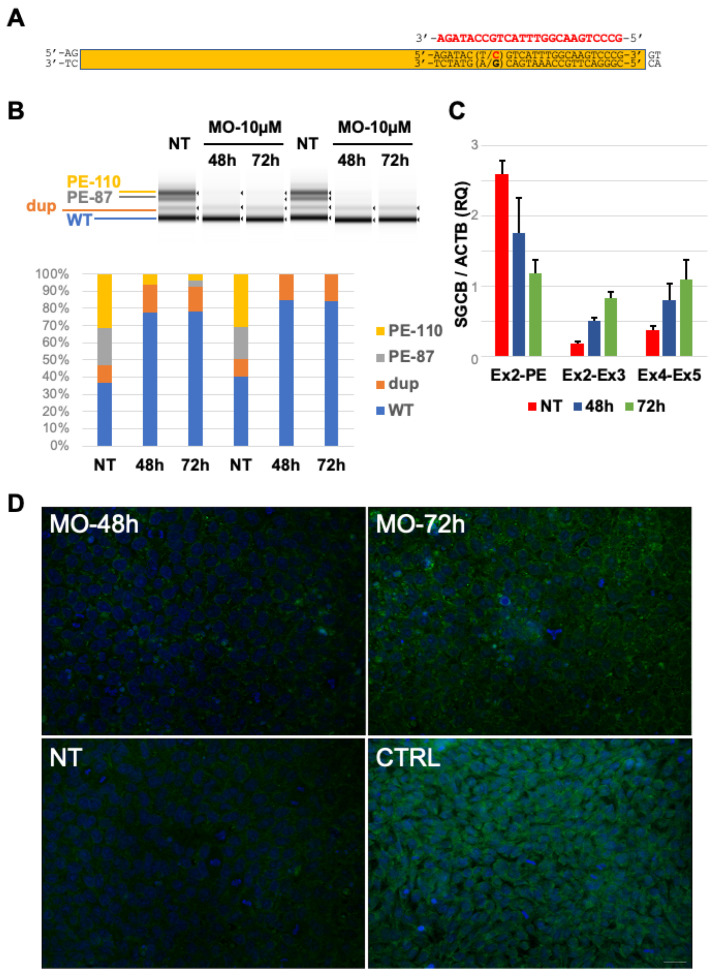
Splicing defect rescue by in vitro morpholino treatment: (**A**) Scheme of the pseudoexon and localization of the Morpholino sequence (MO, in red) used in this study. (**B**) Tape Station analysis of RT-PCR amplicons obtained in untreated (NT) and treated (MO) patient’s derived induced pluripotent stem cells documenting the amelioration of physiological splicing and the reduction in aberrant splicing products after MO delivery. Tape Station analysis allowed the identification of two additional bands corresponding to a predicted shorter form of Pseudoexon (PE-87) and a band with a molecular weight corresponding to the SGCB transcript containing the c.377_384dup duplication (dup). (**C**) qRT-PCR analysis of *SGCB* transcripts encompassing different exon junctions in untreated (NT) patient’s derived iPSCs and in Morpholino (MO)-treated cells displaying the increase in physiological splicing and the concomitant reduction in aberrant transcripts including Exon2-PE junction. (**D**) Immunocytochemical analysis of beta-sarcoglycan signal in untreated patient’s derived iPSCs (NT), in Morpholino (MO)-treated cells, and in iPSCs obtained from a control subject (CTRL). Magnification 40×. Scale bar 50 μm.

## Data Availability

The raw data supporting the conclusions of this article will be made available by the authors, without undue reservation.

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
