# Peer review of "Antisense Morpholino-Based In Vitro Correction of a Pseudoexon-Generating Variant in the SGCB Gene"

_ijms, 2022, doi:10.3390/ijms23179817_

Round 1

Reviewer 1 Report

The authors found a deep intronic mutation which causes a pseudoexon inclusion in patients with beta sarcoglycanopathy. They tried morpholino-based in vitro correction. Though the mutation managed in this article is an ultrarare mutation, this approach has therapeutic implications as it may be expanded to splicing modifications in other variants. However, I have a few concerns.

1. It is advised to describe the pathogenicity of the variants according to the ACMG guideline.

2. Transcript amount of exon 4-5 is much less than expected from nonsense mediated decay of both alleles (Fig. 2E). Transcript amount of exon 2-3 from c.377_384dup is also supposed to be higher (Fig. 2E). On the other hands, the western blot shows a decent amount of beta-sarcoglycan in the patients muscle (Fig. 1D). The authors need to explain these discrepancy in stoichiometry.

3. Description on PE-87 and dup is missing (Fig. 3B).

4. It looks like the antisense morpholino has blocked pseudoexon inclusion from both alleles (Fig. 3B). However, it seems ex2-PE product is still present in the treated cells as half amount of that from the untreated cells (Fig. 3C).

5. Restoration of beta-sarcoglycan is not convincingly strong in Fig. 3D, in contrast to the complete restoration of the transcript (Fig. 3B). It is advised to quantify with western blot.

6. Discussion on the key results of in vitro correction is missing.

Reviewer 2 Report

The authors present a case presenting with LGMDR4 that presented relatively mild, identifying a known class 5 pathogenic variant and a novel intronic variant c.243+1548T>C that causes inclusion of a non-canonical exon, probably through interference with binding of a SRSF2 protein. They show a same phenotype in a previously molecularly unsolved case. In vitro, they show improvement of protein expression following treatment with a morpholino directed against the intronic variant.  

there are some remarks that should should be easily addressed

major remark:

- there is no control morpholino (scrambled morpholino) and no control cell line used in the rescue experiments.

- quantification of the western blots and RT-PCR experiments should be shown with a bar chart. 

- an additional DNA-protein binding experiment would be helpful to support the theory of altered SRSF2 binding 

minor remarks:

- mutation is a term that has been abandoned: use pathogenic or likely pathogenic variant or variant of unkown clinical significance (or class 5, 4 or 3 variant)

- explain LGMDR4 on first use.

- p2, l 74: 'NGS - based analysis followed by trnascript analysis': transcriptome analysis?  or was a cDNA analysis of the SGCB gene done? please specify. 

- p2 l 81: repeated finding of high creatine-kinase levels

-p4,l128: SRSF2 ipv SRFS2

-p4, l131: inherited from the unaffected patients' father and mother, respectively

-p4,l131: The CK levels  in the patients'parents

-p5, l 182: suggest that the c.243+1548T>C variant

-p7,l230_233: segregation is also an important argument for the pathogenecity. also: the independent patient is likely a distant relative based on the haplotype analysis
